# Current Approaches for the Curative-Intent Surgical Treatment of Pancreatic Ductal Adenocarcinoma

**DOI:** 10.3390/cancers15092584

**Published:** 2023-04-30

**Authors:** Maciej Słodkowski, Marek Wroński, Dominika Karkocha, Leszek Kraj, Kaja Śmigielska, Aneta Jachnis

**Affiliations:** 1Department of General, Gastroenterologic and Oncologic Surgery, Medical University of Warsaw, 02-097 Warsaw, Poland; marek.wronski1@wum.edu.pl (M.W.); dominika.karkocha@wum.edu.pl (D.K.); kaja.smigielska@wum.edu.pl (K.Ś.); aneta.jachnis@wum.edu.pl (A.J.); 2Department of Molecular Biology, Institute of Genetics and Animal Biotechnology, Polish Academy of Sciences, 05-552 Jastrzębiec, Poland; 3Department of Oncology, Medical University of Warsaw, 02-097 Warsaw, Poland

**Keywords:** pancreatic cancer, resectability, radical resection, adjuvant treatment

## Abstract

**Simple Summary:**

Pancreatic cancer remains one of the most dreaded cancers worldwide. The incidence of this relatively rare cancer is increasing by almost 1.0% per year. It is estimated that it will become the second leading cause of cancer-related mortality within the next 10 years. Despite the advances in oncology, radical resection, followed by adjuvant systemic chemotherapy, still offers the only realistic chance of curing this disease. Patient selection in specialized and high-volume centers improved perioperative morbidity and mortality rates over the last two decades. Additionally, due to improvements in both surgical techniques and systemic chemotherapy, the indications for resection are expanding to include more locally advanced cases. However, the role of neoadjuvant chemotherapy or radiotherapy remains unclear. This paper summarizes the data regarding current surgical management of pancreatic cancer and reviews future treatment prospects and the latest advances in perioperative strategies.

**Abstract:**

Radical resection is the only curative treatment for pancreatic cancer. However, only up to 20% of patients are considered eligible for surgical resection at the time of diagnosis. Although upfront surgery followed by adjuvant chemotherapy has become the gold standard of treatment for resectable pancreatic cancer there are numerous ongoing trials aiming to compare the clinical outcomes of various surgical strategies (e.g., upfront surgery or neoadjuvant treatment with subsequent resection). Neoadjuvant treatment followed by surgery is considered the best approach in borderline resectable pancreatic tumors. Individuals with locally advanced disease are now candidates for palliative chemo- or chemoradiotherapy; however, some patients may become eligible for resection during the course of such treatment. When metastases are found, the cancer is qualified as unresectable. It is possible to perform radical pancreatic resection with metastasectomy in selected cases of oligometastatic disease. The role of multi-visceral resection, which involves reconstruction of major mesenteric veins, is well known. Nonetheless, there are some controversies in terms of arterial resection and reconstruction. Researchers are also trying to introduce personalized treatments. The careful, preliminary selection of patients eligible for surgery and other therapies should be based on tumor biology, among other factors. Such selection may play a key role in improving survival rates in patients with pancreatic cancer.

## 1. Introduction

Pancreatic ductal adenocarcinoma (PDAC) is the fourth leading cause of cancer-related deaths worldwide. With its rising incidence rates, PDAC is expected to become the second leading cause of cancer-related deaths by 2030 [1,2,3,4]. Despite recent advances in diagnostic tools, chemotherapy, and surgical treatment, the prognosis of PDAC remains poor, with an average 5-year survival rate of 11.5% [5]. The median age of patients at the time of diagnosis is 68 years, and only 6.2% of patients have early-onset disease (diagnosed before the age of 50) [6]. The lack of early alarm symptoms and the late manifestation of the disease are the main reasons why a high number of patients present with locally advanced or metastatic cancer. Radical surgical treatment is the only possible cure for pancreatic cancer (PC) patients, so it is important to carefully qualify patients for a suitable management.

Over the last two decades, more expanded surgical procedures have been introduced in high-volume centers [7]. This was possible due to advances in perioperative care and in the management of complications. The treatment of potentially resectable PDAC is complex, with adjuvant therapy being the standard of care. However, only less than 20% of patients are eligible for upfront radical resection. One well-established protocol for borderline resectable (BR) PC is neoadjuvant treatment (NAT), which makes radical resection possible in a significant proportion of patients [8,9].

Patients with locally advanced (LA) PC usually undergo initial chemotherapy followed by a re-assessment and qualification for either resection or palliative therapy. High volume centers for pancreatic surgery offer more aggressive resection approaches, involving major vascular resection and reconstruction [10]. Metastatic PC is regarded as inoperable, but there is an ongoing discussion, with a growing number of experts acknowledging the beneficial aspects of concomitant radical PC and liver metastasis resection in highly selected patients diagnosed with stage IV disease [11].

PC is an aggressive disease, often with early systemic dissemination. Most of the patients eventually relapse, which suggests the presence of micrometastases at the time of surgery [12,13,14]. This review article summarizes the recent advances in PC surgery.

## 2. Patient Selection

Adequate patient selection, based on tumor morphology and patient-related factors, is a key factor determining the outcome of PC surgery. Cancer staging does not preclude further patient evaluation, as some patients are not fit for major surgery because of comorbidities. Since approximately 70% of PCs are located in the head of the pancreas, radical pancreatoduodenectomy is the procedure of choice despite the high, associated morbidity and mortality.

The recently proposed new criteria for resectability include anatomical, biological, and conditional factors (Figure 1) [15,16]. One biological criterion for BR PC is carbohydrate antigen 19-9 (Ca 19-9) levels of more than 500 U/mL, which is a potential indication for NAT [17].

Prehabilitation in high-risk patients is drawing more and more interest, as the number of elderly, frail patients with multiple comorbidities presenting with technically resectable PC is increasing [18]. Treatment strategy selection (curative or palliative) is the main determinant of prognosis. The median survival in patients eligible for radical resection followed by adjuvant chemotherapy is significantly higher (28 months) in comparison with that after palliative chemotherapy alone (7–11 months), or best supportive care alone (2 months) [19,20,21]; thus, increasing the resection rates is crucial.

Proper patient selection, including the anatomical conditions and tumor biology, as well as an accurate and consistent discrimination between BR and LA PC play a key role in improving treatment outcomes [22]. The definition of resectability for BR/LA PC varies among centers, and it depends mainly on the level of expertise in the use of such complex procedures as vascular resection and reconstruction [23].

### 2.1. Resectable Pancreatic Cancer

Resectable PC is defined as a non-metastatic tumor that does not invade the superior mesenteric artery (SMA), celiac artery, superior mesenteric vein, or portal vein, or has a less than 180° contact with the superior mesenteric vein and/or portal vein, with no vascular contour irregularities.

Although NAT may be implemented in selected cases of resectable tumors, the current data on this course of treatment are inconclusive [24,25]. A recent meta-analysis of six randomized trials [26] showed that NAT in resectable PC did not improve the overall survival (OS) or disease-free survival (DFS), despite increasing R0 resection rates. The role of NAT in this group of patients remains unclear. Decisions regarding the treatment of patients with resectable PC should be made by multidisciplinary teams, and whenever possible, NAT should be conducted as part of prospective clinical trials. The main goal of NAT is to improve the OS by reducing tumor size to facilitate subsequent R0 resection [27]. Furthermore, studies demonstrated that up to 45% of patients are unable to receive adjuvant treatment after radical resection, whereas a fully completed adjuvant chemotherapy regimen is an independent positive prognostic factor for survival. In this context, interest in primary systemic treatment is increasing [28,29,30]. Several observational studies suggest NAT benefits [31,32,33]. The PREOPANC study [34] is the first randomized trial addressing this problem. The PREOPANC-2 trial compares the effects of FOLFIRINOX (triplet chemotherapy with 5-fluorouracil, irinotecan, and oxaliplatin) with those of gemcitabine-based radio-chemotherapy. Additionally, the concept of “total neoadjuvant chemotherapy” has been investigated [35,36]. A German multicenter randomized trial by Kunzmann et al. showed that the effects and safety of nab-paclitaxel plus gemcitabine are similar to those of sequential nab-paclitaxel plus gemcitabine followed by FOLFIRINOX in locally advanced PDAC. About a third of patients in each study arm underwent surgical exploration after induction chemotherapy. The surgical conversion rate with complete macroscopic tumor resection was 35.9% (95% CI 24.3–48.9) in the nab-paclitaxel plus gemcitabine group and 43.9% (31.7–56.7) in the sequential FOLFIRINOX group (odds ratio 0.72 [95% CI 0.35–1.45]; *p* = 0.38). At two years’ follow-up, the median OS was comparable (18.5 months vs. 20.7 months; hazard ratio 0.86 [95% CI 0.55–1.36]; *p* = 0.53) [37]. Recently, a Japanese randomized trial found a significant survival benefit of neoadjuvant chemotherapy with gemcitabine and S1 compared with upfront surgery in patients with resectable and BR tumors with portal vein infiltration. In this study, the median OS was 36.7 months in the neoadjuvant arm in comparison with 26.6 months in the upfront surgery arm (HR 0.72 [95% CI 0.55–0.94]; *p* = 0.015) [38]. However, there are still not enough studies on this subject.

Upfront surgery with radical resection is still recommended for most patients with adequate performance status, no major comorbidities, and low CA 19-9 levels. The benefits of upfront surgery with adjuvant chemotherapy may outweigh those of NAT. First, biliary stenting for obstructive jaundice can be omitted in certain groups of patients. Second, patients avoid the risk of their general condition deteriorating during chemotherapy and the risk of disease progression if the tumor is not sensitive to chemotherapy [39]. The goal of the primary operation is radical R0 resection. It is unreasonable to perform non-radical resection (R2), as the outcomes are similar to those in patients who do not undergo surgery.

Despite undergoing a curative-intent surgery, a majority of patients develop local recurrence or distant metastases and eventually die within 2 years of the operation. The overall recurrence rate is approximately 85%, and the 5-year survival is less than 30% [13,30]. The quality of primary resection plays a key role in improving survival.

PC resection requires the primary tumor to be removed with the surrounding tissues and lymph nodes. The anatomical position of PC relative to the planes of surgical dissection is the reason why a majority of removed tumors have cancer cells detected within 1 mm of the resection margin (R1 resection), despite an optimal surgical technique. Hence, the reported R1 resection rates range from 28% to 71% [40,41,42,43]. The choice of the type of pancreatic resection is determined by the primary tumor location. Pancreatoduodenectomy is performed in the case of tumors found in the head of the pancreas, while distal pancreatectomy is used for tumors located in the pancreatic body and tail. Whenever partial pancreatectomy might result in a positive resection margin, total pancreatectomy should be performed instead. This type of resection is also used to prevent life-threatening pancreatic fistulae after pancreatoduodenectomy in patients with a fatty, soft, and fragile pancreas. Patients with resectable tumors should undergo upfront surgery, while BR tumors should be managed with NAT [39].

The biology of PDAC is characterized by aggressive local growth, early regional lymph node involvement, and perivascular nerve and lymphatic vessel invasion. Hence, radical pancreatic resection should address three critical issues: an adequate resection margin, and adequate lymph node and visceral blood vessel clearance (Figure 2).

Tumor-free resection is achieved by a wide excision of the pancreatic tumor, confirmed via intraoperative frozen section microscopy assessing the pancreatic and biliary margins. Currently, pancreatoduodenectomy involves resection of the head of the pancreas combined with total excision of the so-called mesopancreas [44]. The TRIANGLE procedure involves vessel-oriented dissection and the removal of all soft tissues found in the triangular space bordered by the celiac trunk cranially, the SMA caudally and the portal vein anteriorly [45].

Mesopancreas dissection is divided into three levels and allows complete excision of the perivascular tissue [46]. Level 1 dissection removes the mesopancreas while sparing the lymphatic and nervous structures along the SMA. Level 2 dissection involves removing the lymph nodes located close to the SMA, and level 3 dissection clears the SMA from nervous and lymphatic structures. Level 3 dissection is recommended in PC patients. In contrast, arterial divestment is a procedure performed in patients after NAT with confirmed peripancreatic artery involvement without arterial wall infiltration. During this procedure, the dissection plane may be similar to that used in level 3 mesopancreas dissection, which spares the adventitial layer, or it may follow a deeper plane between the adventitia and the external elastic lamina [47]. In selected cases, arterial wall infiltration is an indication for partial resection of the artery, otherwise pancreatectomy should be abandoned. Although imaging-based assessment of arterial wall infiltration after NAT is difficult, arteries with ≤270° circumferential and <26 mm longitudinal contiguity with solid soft tissue are considered unlikely to be invaded [48].

One important step during a pancreatoduodenectomy procedure is the gastroduodenal artery (GDA) clamping test, which is used to exclude severe and clinically significant stenosis of the celiac artery, which precludes safe resection of the pancreatic head. A positive GDA clamping test necessitates releasing the median arcuate ligament first, and then—if the flow in the common hepatic artery (CHA) does not improve—arterial reconstruction.

Standard lymphadenectomy associated with pancreatoduodenectomy includes the following lymph node stations: 5 (suprapyloric), 6 (infrapyloric), 8a (anterior along the CHA), 12b1 (along the common bile duct), 12c (around the cystic duct), 13 (posterior aspect of the pancreatic head), 14a and 14b (right lateral side of the SMA), 17 (anterior aspect of the pancreatic head) [49]. Hepatoduodenal ligament dissection should be performed up to the level where the right hepatic artery traverses to the right hepatic lobe. Standard lymphadenectomy should provide at least 15 lymph nodes for appropriate pathologic staging. Extended lymphadenectomy was not demonstrated to provide any better long-term prognosis in pancreatic adenocarcinoma [50,51].

Currently, pancreatoduodenectomy often involves the artery-first approach. The upfront dissection of the SMA ensures early detection of possible arterial involvement and helps avoid incomplete resection. Several approaches to the SMA have been described [52], with the right posterior approach being the most common. However, a combination of different approaches is usually necessary. Infiltration of the portal or superior mesenteric vein requires en-bloc resection, which is facilitated by the artery-first approach.

The International Study Group of Pancreatic Surgery (ISGPS) recognizes four types of vein resection and reconstruction [53]. Type 1 involves tangential resection with direct venorrhaphy. Type 2 is a wedge venous resection that requires patch placement. Types 3 and 4 are segmental resections, either with an end-to-end anastomosis reconstruction or vascular graft interposition, respectively. Additional procedures at the time of pancreatoduodenectomy might negatively influence the postoperative course and outcomes.

Recently, pancreatoduodenectomy was classified into four categories with increasing mortality and morbidity rates [54]: standard pancreatoduodenectomy (type 1), pancreatoduodenectomy combined with vein resection (type 2), pancreatoduodenectomy with multi-visceral excision (type 3), and pancreatoduodenectomy with arterial resection (type 4). Type 4 procedures are of the greatest risk and are associated with the worst prognosis. The most serious complication of pancreatoduodenectomy is postoperative pancreatic fistula, which significantly increases morbidity and mortality. As of this moment, none of the techniques of pancreatic anastomosis or mitigation strategies has been shown to significantly decrease the rate of pancreatic fistulas [55].

When the body and tail of the pancreas are involved, radical tumor excision requires adequate removal of retroperitoneal tissues. This is ensured by radical antegrade modular pancreatosplenectomy (RAMPS). The procedure begins with early ligation of the splenic vein and artery at their origins. The left side of the SMA is cleared of all the nerve and lymphatic tissue (level 3 dissection), followed by medial-to-lateral dissection along the plane deep to the anterior renal fascia (anterior RAMPS) or behind the left adrenal gland (posterior RAMPS). Standard lymphadenectomy in a distal pancreatectomy includes the removal of three lymph node stations: 10 (in the splenic hilum), 11 (along the splenic artery), and 18 (along the inferior margin of the pancreatic body and tail) [49]. The number of examined lymph nodes is particularly important for PC staging. Several studies showed that an adequate number of lymph nodes is associated with a better prognostic assessment after curative resection, but the optimal cut-off number of lymph nodes has not been established. Some authors advocate for a range from 11–17 to 19, with a minimum of 11 lymph nodes required to obtain satisfactory accuracy of tumor staging [56].

### 2.2. Borderline Resectable Cancer

There are several definitions of BR PC, including the international consensus criteria. BR PC is usually defined as neither clearly resectable nor unresectable disease that requires downstaging to achieve R0 resection (Figure 3) [57]. Patients with BR PC have no evidence of metastatic disease but are less likely to undergo R0 resection because of venous and arterial involvement. The goal of treatment is to maximize the chances of negative-margin resection. Typically, BR PC patients receive NAT, and those who do not have distant progression or local invasion precluding surgery subsequently undergo surgical exploration and resection [23]. BR PC patients receive NAT for 2 to 6 months before surgery. The goal of therapy is to improve resectability by downstaging the primary tumor, reducing the number of micrometastases, and helping avoid surgery in patients with aggressive metastatic biology [58]. Boone et al. demonstrated that the serum CA 19-9 response to FOLFIRINOX- or gemcitabine-based NAT in BR PC patients helped predict an R0 resection [59]. The only randomized trial supporting NAT in BR PC was the PREOPANC-1 study [34]. The study population were patients with both resectable and BR disease. The authors reported improved OS and DFS rates if gemcitabine and chemoradiotherapy were used prior to surgery in comparison with those rates following an upfront resection. The 2020 National Comprehensive Cancer Network (NCCN) guidelines recommend NAT for BR PC patients [60]. Preoperative treatment has been associated with several potential benefits including tumor shrinking with decreased nodal involvement, increased margin-negative resection rates, early treatment of occult micrometastases, improved compliance with chemotherapy, improved survival after curative resection, and better selection of patients who were more likely to benefit from surgery [30].

The management of patients with BR PC eligible to receive NAT is similar in most centers, although in the absence of level-1 data, there is a lack of consensus regarding the optimal timing and sequencing of treatment [23]. A recent meta-analysis of seven randomized controlled trials (RCTs) confirms the superiority of NAT in patients with BR PC [39]. In all seven RCTs the NAT regimen included gemcitabine without nab-paclitaxel. Only one of the four arms of the ESPAC-5F study included 20 patients with neoadjuvant FOLFIRINOX administration [61]. In this study, there was no significant difference in resection rates between patients who underwent upfront surgery and those who received NAT. Short-course (8-week) NAT had a significant survival benefit over upfront surgery. Neoadjuvant chemotherapy with either gemcitabine–capecitabine combination or FOLFIRINOX showed higher survival rates in comparison with upfront surgery. These findings support the use of short-course neoadjuvant chemotherapy in patients with BR PC. Venous resection is often necessary in BR PC in order to achieve clear surgical margins and acceptable oncological outcomes. Currently, approximately 4–20% of pancreatoduodenectomy procedures involve venous resection. From an oncological point of view, similar OS rates have been reported after resection with or without veins, which means that tumor invasion is not associated with particular aggressiveness but only with location [62]. When a part of venous circumference is involved, but the lumen remains patent, traditional vein reconstruction can be performed in a number of ways. Usually, complete mobilization of the specimen, including dissection of the SMA, is accomplished. This leaves the specimen attached only at the site of venous involvement. Proximal and distal control of the portal or superior mesenteric vein is obtained with vascular clamps and the vein is resected en bloc with the specimen. Venous reconstruction can be accomplished via several techniques.

In the case of sidewall adherence, a longitudinal ellipse of the vein can be resected, and the vein can be closed with transverse suturing or with a patch. For segmental resection of shorter segments (<3–4 cm) primary end-to-end anastomosis can be performed to allow tension-free reconstruction. When longer segments are involved, the use of an interposition graft is preferred with native veins (the internal jugular, renal, or superficial femoral vein) [62,63,64,65]. To improve outcomes, vascular resections should be performed at high-volume centers by surgeons with experience in performing such procedures.

### 2.3. Locally Advanced Pancreatic Cancer

LA PC accounts for 30% of newly diagnosed cases and is considered surgically unresectable due to local involvement of the adjacent critical blood vessels [66]. Generally, LA PC is considered incurable.

The standard of care is very similar to that used for patients with metastases, and it involves at least 6 months of chemotherapy [67]. The surgical outcomes of pancreatectomy performed in combination with vascular reconstruction have improved recently, and as neoadjuvant combination regimens are becoming more effective, there is a growing interest in identifying those patients with LA PC who may benefit from a more aggressive surgical approach.

The TRIANGLE operation was initially described as a method of radical resection after NAT in LA PC [10]. A proportion of 10–20% of patients who received a full course of chemotherapy are eligible for radical resection. Current imaging techniques do not help differentiate between tumor and fibrotic tissues around arterial structures, such as the celiac artery, CHA, and SMA. The goal of the TRIANGLE procedure is to remove all tissues from around the vessels and demonstrate the absence of viable tumor via intraoperative frozen section microscopy. The procedure usually involves simultaneous major vein resection and reconstruction. In patients after NAT this approach has been described as the “periarterial divestment” technique [47]. This technique aims at radical tumor clearance without arterial resection and is characterized by entering the adventitial layer between the arterial wall and remnant tumor/fibrotic tissue. Once this layer is opened, it is the guiding plane for dissection, which allows the surgeon to avoid arterial resection. It is essential that the arteries are dissected along the adventitial layer, as this allows for complete lymphadenectomy and soft tissue removal from the respective area [7]. Dissection of lymphatic tissues is continued to the origins of the celiac artery and SMA. This radical approach results in the visualization of an anatomic triangle, cleared from surrounding tissues, bordered by the portal vein, the SMA, and the celiac artery. Besides its use in LA PC, the TRIANGLE operation can, and potentially should, be performed in all resectable and BR tumors in order to achieve a truly radical surgery [68,69]. However, there are certain limitations for resection, such as arterial involvement. Resection of the celiac artery, the CHA, or the SMA is associated with a significant risk of serious complications and death [70]. Mostly due to aforementioned concerns, arterial resection is avoided, which results in unresectability or a non-radical resection [45]. One should bear in mind that the TRIANGLE operation does not resemble any other type of extended lymphadenectomy. Instead, it focuses on the site where the microscopic tumor spread is most commonly observed and on the ‘hot spots’ of frequent tumor recurrence. Other approaches have usually aimed at the removal of not only local lymph nodes but additionally those lymph nodes located in the interaortocaval space. Such an extended lymphadenectomy is not recommended, as it failed to improve survival and was often associated with an increased postoperative morbidity. In contrast, the TRIANGLE operation may lead to improved local radicality and reduced local recurrence [70]. However, the impact of local radicality achieved by the TRIANGLE operation and arterial divestment on overall survival still awaits confirmation. If resection of a venous segment followed by venous reconstruction is feasible, cancers involving the portal vein and/or superior mesenteric vein can still be removed.

Due to the high morbidity and mortality associated with pancreatoduodenectomy combined with complex vascular reconstruction, careful patient selection is needed. Management of such cases additionally requires significant clinical expertise [7,23].

### 2.4. Metastatic Disease

Approximately 50% of pancreatic cancer patients already have distant metastases at the time of diagnosis. For those who undergo curative resection, over 70% will have disease recurrence, with secondary lesions mostly found in the liver, lungs, or peritoneum [71].

Metastatic (stage IV) PC is considered only for palliative treatment, and surgical intervention is contraindicated. The current standard treatment is palliative chemotherapy. The role of hepatectomy in patients with liver metastases remains controversial [72,73]. However, there are several publications supporting a more aggressive approach involving curative-intent surgery with the resection of the primary tumor and metastases [11,74,75]. Simultaneous resection of PDAC and liver metastases could be beneficial in selected patients. Hackert et al. have published the largest series of articles on PDAC resection to date [76]. Those authors resected 85 liver metastases, with a median survival of 12.3 months and a 5-year survival of 5.9%. Andreou et al. evaluated postoperative outcomes and long-term survival in patients after combined pancreatic and liver resection for synchronous liver metastases. They reported postoperative morbidity and mortality rates of 50% and 5%, respectively; 1-, 3-, and 5-year OS rates of 41%, 13%, and 7%, respectively; and 1-, 3-, and 5-year DFS rates of up to 39%, 9%, and 5%, respectively [73].

Resection of metastases located in the lungs could be beneficial in a selected group of patients. Isolated metachronous lung metastases in PDAC patients are well known to be associated with a better prognosis than metastases to other sites [62].

The most important factors influencing OS following synchronous metastasis resection are an R1 margin status at liver resection, a T4-stage tumor, regional lymph node involvement, poorly differentiated cancer, and the absence of pre- or postoperative chemotherapy. Therefore, perioperative adjuvant treatment modalities may be crucial to improve survival. Such treatment must be customized and should be offered only in highly specialized centers [73,77].

In their conclusions to a recently published review on this topic, Sakaguchi et al. suggested a substantial survival benefit in patients with synchronous metastases who responded favorably to initial chemotherapy [75]. It is necessary to establish qualification criteria for surgical resection for patients with metastatic PDAC.

### 2.5. Treatment of Recurrent Disease

Traditionally, local recurrence of PC after radical resection is recognized as a condition suitable only for palliative management, with no indication for surgical treatment. However, some local lesions may be treated with re-resection. The aim of the procedure remains the same as in primary tumor resection. Only radical (R0) resection can improve survival. Aggressive chemotherapy regimens are introduced to stabilize recurrent disease and increase the possibility of a curative-intent surgical intervention [78]. The patients considered as potential candidates for this management should undergo intense chemotherapy before re-resection. According to Nienhuser et al., re-resection of recurrent PC offers a significant survival benefit in selected patients, with acceptable procedure-related morbidity and mortality [79]. Clinical parameters associated with an improved prognosis after local recurrence of PC are patient age (<65 years), good performance status, and the time from initial resection of >10 months. Due to their prognostic significance, molecular markers, such as KRAS and SMAD4, could potentially further improve patient selection.

### 2.6. Adjuvant Therapy

Surgical resection is the basis of radical treatment and remains the only way to a complete cure in PDAC patients. However, despite radical surgical resections, the postoperative recurrence rate is very high, with nearly 90% of patients without any adjuvant treatment experiencing a recurrence, mainly in the form of distant metastases. Optimization of postoperative treatment over the last two decades, especially in the field of postoperative chemotherapy, increased the 5-year survival rate to about 30% [80].

The first large, randomized study that showed for the first time the positive effect of postoperative treatment on OS was the ESPAC-1 study (the European Study Group for Pancreatic Cancer). In this four-arm study, there was a significantly increased survival in groups receiving a fluorouracil-based adjuvant chemotherapy in comparison with those who did not (the median OS of 20.1 vs. 15.5 months, *p =* 0.009). No additional benefits of chemoradiotherapy over chemotherapy alone were reported (the median OS of 15.9 vs. 17.9 months, respectively) [81].

Another study that demonstrated the effects of 6-month adjuvant chemotherapy was the CONKO-001 trial (Charite Onkologie). This study compared the use of gemcitabine monotherapy with the use of no adjuvant treatment. The benefit of adjuvant treatment in terms of OS was observed (a median OS of 22.8 vs. 20.2 months). Adjuvant treatment also improved the 5-year and 10-year survival rates (improvement from 10.4% to 20.7% and from 7.7% to 12.2%, respectively) [82]. The ESPAC-3 study directly compared the effectiveness of adjuvant gemcitabine-based chemotherapy with a fluorouracil-based regimen. There was no difference in OS (a median OS of 23.6 vs. 23.0 months, *p* = 0.4), demonstrating an equivalence of those two regimens. However, differences in toxicity and serious adverse event rates were noted in favor of gemcitabine (14% with fluorouracil vs. 7% with gemcitabine, *p <* 0.01) [83].

The subsequent ESPAC-4 study, compared gemcitabine monotherapy with a combination therapy of gemcitabine and capecitabine (GEMCAP). In this study, the two-drug regimen showed superiority in terms of OR (the median OS of 28.0 vs. 25.5 months, *p =* 0.03). GEMCAP is the only gemcitabine doublet that has conclusively demonstrated efficacy as adjuvant therapy [20]. Combinations of gemcitabine with erlotinib (CONKO-005-trial) or nab-paclitaxel (APACT-trial) did not prove to be any more effective as adjuvant treatment than gemcitabine monotherapy, with both combination regimens failing to demonstrate a benefit in DFS [84,85].

A significant breakthrough in the adjuvant treatment of PDAC came with the use of the multi-drug chemotherapy regimen mFOLFIRINOX (modified 5-fluorouracil, irinotecan, and oxaliplatin). The PRODIGE 24/CCTG PA.6 randomized phase III trial evaluated the effectiveness of a 6-month mFOLFIRINOX regimen compared with gemcitabine monotherapy [21]. The study included 493 patients, aged under 79 years, with a good Eastern Cooperative Oncology Group (ECOG) performance status (0–1, and eligible for adjuvant chemotherapy within a period of less than 12 weeks after macroscopic radical resection. The primary endpoint in this study was DFS. After a median follow-up of 33.6 months, the median DFS was 21.6 months in the mFOLFIRINOX group and 12.8 months in the gemcitabine group (*p <* 0.001). Moreover, the median OS was 54.4 months in the mFOLFIRINOX group, compared with 35.0 months in the gemcitabine group (*p =* 0.003). These are the best results for any adjuvant therapy to date. Notably, the mFOLFIRINOX regimen has a higher risk of side effects (76% of patients developed grade 3–4 toxicity in the mFOLFIRINOX arm vs. 51% in the gemcitabine arm). The planned 6-month adjuvant treatment was completed by 66% of patients in the mFOLFIRINOX group, compared with 79% in the gemcitabine monotherapy arm.

In conclusion, adjuvant chemotherapy based on the mFOLFIRINOX regimen is currently the established standard of adjuvant treatment for fit patients. For elderly patients with a poorer performance status (ECOG-2), gemcitabine monotherapy or a gemcitabine-capecitabine combination remains a therapeutic option. However, not all patients after surgery are able to use adjuvant treatment due to postoperative complications and significant comorbidities.

## 3. Conclusions

This century has seen a significant improvement in PC treatment. This is mainly due to earlier diagnoses, higher numbers of radical surgeries, and advances in neoadjuvant and adjuvant therapy. Data suggest an improved OS even in metastatic disease. Radical surgery with adequate management of complications results in low mortality rates and plays a key role in multimodal treatment of PDAC. Surgery remains the only curative treatment for PDAC. Pancreatic surgery centralization results in decreased perioperative mortality and prolonged OS. NAT, whose impact on BR and LA tumors is unquestionable, may increase the chances of R0 resection in these patients. There are more data regarding the efficacy of NAT in resectable PC. Moreover, there are ongoing studies that aim to investigate this issue further, thus, new data are expected in the nearest future. Nevertheless, further studies on how to improve prevention, early diagnosis, and better understand the molecular biology of PC should lead to the development of more effective systemic treatments and better surgical outcomes. In order to introduce personalized therapy in PDAC, the specific genetic mutations must be assessed first. It is also important to bear in mind that the multidisciplinary care of patients with PDAC should also involve improving the quality of life.

## Figures and Tables

**Figure 1 cancers-15-02584-f001:**
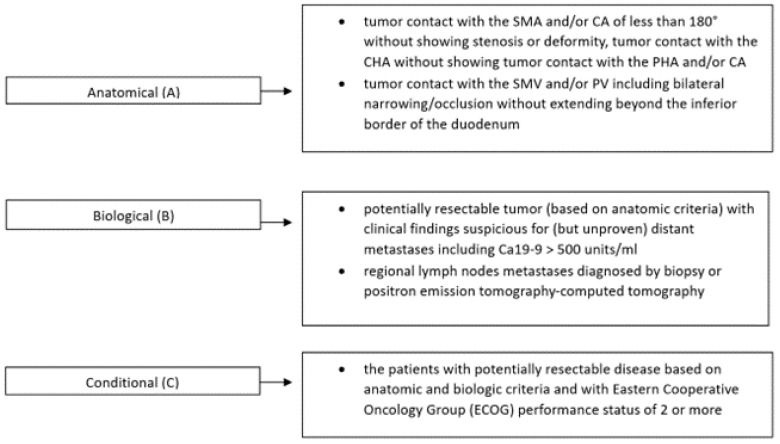
Definition of patients with BR-PDAC according to the three distinct dimensions: anatomical (A), biological (B), and conditional (C), proposed by the international consensus on definition and criteria of borderline resectable PDAC (2017) [15]. Abbreviations: SMV, superior mesenteric vein; PV, portal vein; SMA, superior mesenteric artery; CA, celiac artery; CHA, common hepatic artery; PHA, proper hepatic artery.

**Figure 2 cancers-15-02584-f002:**
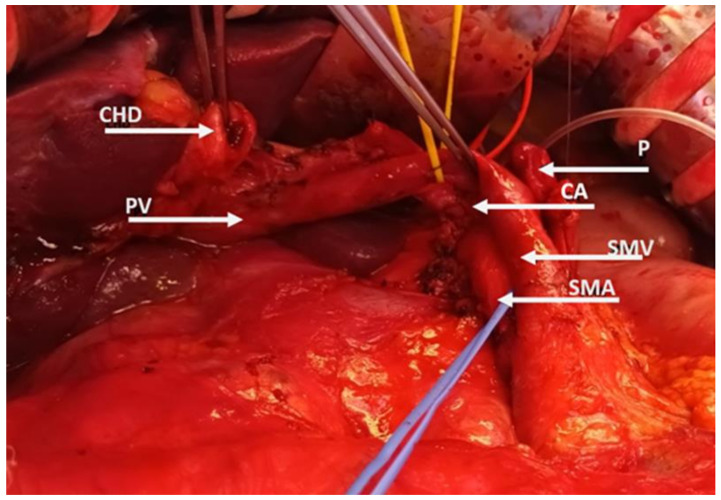
Radical resection with vascular clearance. Abbreviations: SMV, superior mesenteric vein; PV, portal vein; SMA, superior mesenteric artery; CA, celiac artery; CHA, common hepatic artery; PHA, proper hepatic artery; P, pancreas.

**Figure 3 cancers-15-02584-f003:**
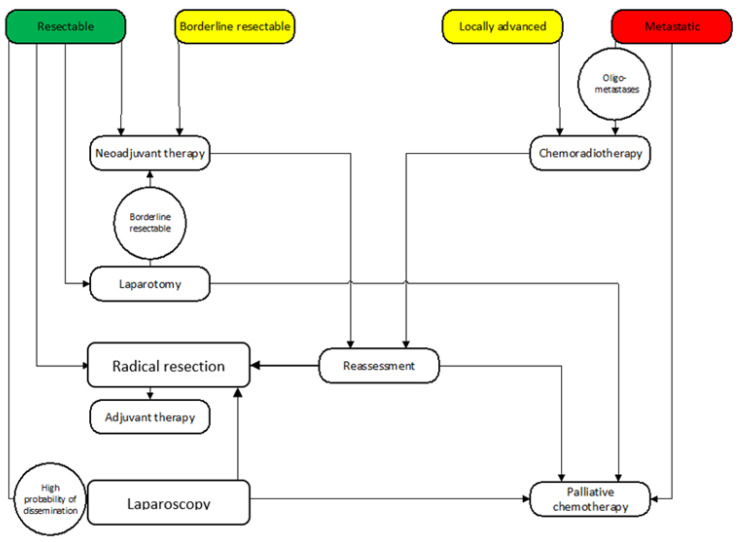
Flow chart of pre- and intraoperative decision making (green color; resectable tumour, yellow; borderline resectable or locally advanced with radical resection after oncological therapy, red; metastatic).

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
