# Peer review of "Current Approaches for the Curative-Intent Surgical Treatment of Pancreatic Ductal Adenocarcinoma"

_cancers, 2023, doi:10.3390/cancers15092584_

Round 1

Reviewer 1 Report

This is a comprehensive review of curative-intent therapy in PDAC. Although, there is not much novelty compared to other reviews on this topic, the manuscript is generally well-written with only minor English language shortcomings. I have only a few comments, which should be addressed.

- in my opinion, the title does not adequately reflect the content. The focus of this manuscript is not "eligibility assessment for radical resection" but is rather more comprehensive on "curative-intent treatment approaches for PDAC"

- figure 1 is in my opinion not helpful at all. What do the authors want to display with this? I would rather suggest to display the actual consensus definition of BR PC (adopted from Isaji et al.)

- figure 2: the black text can be hardly read. I would suggest changing the colour to white. Furthermore, this figure needs a legend.

- figure 3: it remains unclear what you suggest to do if laparoscopy in patients with resectable PDAC but "high probability of dissemination" does not show evidence of metastases?

- Pg. 7: you write that "There is no exact definition of borderline resectable PC". However, there are several "exact" definitions and even an international consensus definition of BR PC (Isaji et al.)

- for the sake of completeness, I would suggest that you include and discuss the trial by Motoi et al. (Jpn J Clin Oncol 2019) on neoadjuvant gemcitabin + S1 and the NEOLAP trial by Kunzmann et al. (Lancet Gastroenterol Hepatol 2021) in the section on neoadjuvant therapy.

- I would recommend to temper the statements regarding the triangle procedure and periarteriel divestment since there is no high-level evidence to support these. In fact, both, the study by Jang et al (Ann Surg 2014) an Lin et al (Cancer Communications 2022) are very similar to the triangle procedure and did not show an improvement of OS.

Author Response

Dear Editors,

Thank you for considering our manuscript “ Eligibility assessment for radical resection in pancreatic cancer” for publication in Cancers. First of all, we greatly appreciate the Reviewers’ comments on our manuscript.  We have considered all their valuable suggestions and made the necessary corrections. We hope the revised version of the manuscript is now acceptable for publication.

            In response to the Reviewers’ comments, the following corrections and modifications have been made in the manuscript:

Reviewer, point 1. “in my opinion, the title does not adequately reflect the content. The focus of this manuscript is not "eligibility assessment for radical resection" but is rather more comprehensive on "curative-intent treatment approaches for PDAC"

Response 1.: We have decided to modify the tittle “Current approaches for the curative-intent surgical treatment ofr pancreatic ductal adenocarcinoma”

Reviewer, point 2. “ (Figure 1)in my opinion not helpful at all. What do the authors want to display with this? I would rather suggest to display the actual consensus definition of BR PC (adopted from Isaji et al.)”

Response 2.: We removed the original version of the figure nr 1 and developed a new one:

Reviewer, point 3. “ figure 2: the black text can be hardly read. I would suggest changing the colour to white. Furthermore, this figure needs a legend.”

Response 3.: We have changed the colours on the graphics and have added a legend.

Reviewer, point 4.” figure 3: it remains unclear what you suggest to do if laparoscopy in patients with resectable PDAC but "high probability of dissemination" does not show evidence of metastases?”

Response 4.: We have added an "arrow" from the laparoscopy, to radical resection which makes the diagram easier to understand.

Reviewer, point 5.” Pg. 7: you write that "There is no exact definition of borderline resectable PC". However, there are several "exact" definitions and even an international consensus definition of BR PC (Isaji et al.)”

Response 5.: We have modified the sentence to: There are several definitions of BR PC, including the international consensus criteria.”

Reviewer, point 6. “for the sake of completeness, I would suggest that you include and discuss the trial by Motoi et al. (Jpn J Clin Oncol 2019) on neoadjuvant gemcitabin + S1 and the NEOLAP trial by Kunzmann et al. (Lancet Gastroenterol Hepatol 2021) in the section on neoadjuvant therapy”

Response 6.: We have added a section of the text including above studies, in the section of neoadjuvant therapy.” A German multicenter randomized trial by Kunzmann et al. showed that the effects and safety of nab-paclitaxel plus gemcitabine are similar to those of sequential nab-paclitaxel plus gemcitabine followed by FOLFIRINOX in locally advanced PDAC. About 1/3 of patients in each study arm underwent surgical exploration after induction chemotherapy. The surgical conversion rate with complete macroscopic tumor resection was 35.9% (95% CI 24.3–48.9) in the nab-paclitaxel plus gemcitabine group and 43.9% (31.7–56.7) in the sequential FOLFIRINOX group (odds ratio 0.72 [95% CI 0.35–1.45]; p=0.38). At two years’ follow-up, the median OS was comparable (18.5 months vs. 20.7 months; hazard ratio 0.86 [95% CI 0.55–1.36]; p=0.53). Recently, a Japanese randomized trial found a significant survival benefit of neoadjuvant chemotherapy with gemcitabine and S1 compared with upfront surgery in patients with resectable and BR tumors with portal vein infiltration. In this study, the median OS was 36.7 months in the neoadjuvant arm in comparison with 26.6 months in the upfront surgery arm [HR 0.72 (95% CI 0.55–0.94; p=0.015] [37]. Furthermore, the results from Kunzmann et al. study suggest that the efficacy and safety of nab-paclitaxel plus gemcitabine are similar to those of nab-paclitaxel plus gemcitabine followed by FOLFIRINOX in PDAC [38]. Preliminary results arouse expectations for better outcomes in PC patients. However, there are still not enough studies on this subject.”

Reviewer, point 7.” I would recommend to temper the statements regarding the triangle procedure and periarteriel divestment since there is no high-level evidence to support these. In fact, both, the study by Jang et al (Ann Surg 2014) an Lin et al (Cancer Communications 2022) are very similar to the triangle procedure and did not show an improvement of OS.”

Response 7.: We supplemented the text with the following sentence: However, the impact of local radicality achieved by the TRIANGLE operation and arterial divestment on overall survival still awaits confirmation.

The details of the revisions to the manuscript:

  1. We have modified the Title
  2. We have modified surnames of the authors (Polish version) in affiliation
  3. We have carried out extensive correction of the English language
  4. We have removed the original version of figure 1 and developed a new one (with the legend), in accordance with the reviewer's suggestions
  5. We have changed the colours on figure 2 and have added a legend
  6. We have added an "arrow" from the laparoscopy, to radical resection which makes the diagram easier to understand (Figure 3)
  7. We have added a few sentences discussing two studies on the role of neoadjuvant therapy in PC patients to subsection 1. Resectable pancreatic cancer
  8. We have also added one sentence to subsection 2. Borderline resectable cancer
  9. We have revised the literature in accordance with the reported studies.

With kind regards,

Manuscript Authors

Reviewer 2 Report

Comments on:

Eligibility assessment for radical resection in pancreatic cancer

I have to congratulate the authors with this thorough review on an important topic. Pancreatic cancer remains one of the few with no real progress in systemic treatment, consequently new developments in surgery are highly valuable.

I have no critical comments for the body text; however I would suggest the authors to revise the title of the manuscript. The presented review covers a much wider subject than just surgery, hence the present title is somehow misleading.

Comments on:

Eligibility assessment for radical resection in pancreatic cancer

I have to congratulate the authors with this thorough review on an important topic. Pancreatic cancer remains one of the few with no real progress in systemic treatment, consequently new developments in surgery are highly valuable.

I have no critical comments for the body text; however, I would suggest for the revision of the title of the manuscript. The presented review covers much wider subject than just surgery, hence the present title is somehow misleading.

Author Response

(The authors gave the same response as above.)
